# Global risk mapping of highly pathogenic avian influenza H5N1 and H5Nx in the light of epidemic episodes occurring from 2020 onwards

**Marie-Cécile Dupas[1,2]\*[†], Maria F Vincenti-Gonzalez[1†], Madhur Dhingra[3], Claire Guinat[4], Timothée Vergne[4], William Wint[5], Guy Hendrickx[6], Cedric Marsboom[1,6], Marius Gilbert[1], Simon Dellicour[1,7,8]**

[1]Spatial Epidemiology Lab (SpELL), Université Libre de Bruxelles (ULB), Brussels, Belgium; [2]Data Science Institute, University of Hasselt, Hasselt, Belgium; [3]Food and Agriculture Organization of the United Nations, Rome, Italy; [4]Interactions Hôtes-Agents Pathogènes (IHAP), Université de Toulouse, Toulouse, France; [5]Environmental Research Group Oxford Ltd, c/o Department of Biology, Oxford, United Kingdom; [6]Avia-GIS research department, Avia-GIS, Zoersel, Belgium; [7]Department of Microbiology, Immunology and Transplantation, Rega Institute, Leuven, Belgium; [8]Interuniversity Institute of Bioinformatics in Brussels, Université Libre de Bruxelles, Vrije Universiteit Brussel, Brussels, Belgium

\*For correspondence: dupas.mc@gmail.com

[†]These authors contributed equally to this work

## eLife Assessment

This global study compares environmental niche model outputs of avian influenza pathogen niche constructed for two distinct periods, and uses differences between those outputs to suggest that the changed case numbers and distribution relate to intensification of chicken and duck farming, and extensive cultivation. While a **useful** update to existing niche models of highly pathogenic avian influenza, the justification for the use of environmental niche models to explore land cover change as a driver of changed case epidemiology is **incomplete**.

**Abstract** Avian influenza (AI) is a highly contagious viral disease affecting poultry and wild water birds, posing significant global challenges due to its high mortality rates and economic impacts. Highly pathogenic avian influenza (HPAI) outbreaks, particularly those caused by H5N1 and its variants, have surged since 1959. The HPAI H5N1 clade 2.3.4.4b viruses have notably expanded their geographical reach, affecting numerous countries, diverse avian species, and now mammals. Using an ecological niche modelling approach, this study aims to elucidate the environmental factors associated with increased HPAI H5 cases since 2020, investigate potential shifts in ecological niches, and predict new areas suitable for viral circulation. We developed ecological niche models for HPAI cases in wild and domestic birds across two distinct periods: 2015–2020 and 2020–2022. Key environmental predictors include chicken and duck population density, human density, distance to water bodies, and land cover variables. Post-2020, we observe increased relative influence of predictors such as intensive chicken population density and cultivated vegetation. Risk maps reveal notable ecological suitability for HPAI H5 circulation in Europe, Asia, and the Americas, with significant expansions of at-risk areas post-2020. Wild bird H5 occurrences appear primarily correlated with urban areas and open water regions. Our analyses also highlight a potential shift in affected wild bird species diversity, with more avian species, particularly sea birds, impacted post-2020. Overall,

these results further contribute to the understanding of HPAI epidemiology and identify regions where surveillance and control measures should be prioritised.

## Introduction

Since 2020, an increase in both H5Nx and H5N1 cases has been observed (*Figure 1*). A variant of the Gs/Gd H5N1 viruses belonging to the H5 clade 2.3.4.4b has led to an unprecedented number of deaths in wild birds and poultry in many countries in Africa, Asia, and Europe (*Huang et al., 2023*). In 2021, the virus spread to North America, and in 2022, to Central and South America (*FAO et al., 2023*) spilling over into poultry farms and infecting an alarming number of wild terrestrial, marine, and domestic mammals (*Elsmo et al., 2023*; *Leguia et al., 2023*; *Neumann and Kawaoka, 2024*; *Plaza et al., 2024*; *Peacock et al., 2024*), marking an unprecedented expansion in the geography and impact of highly pathogenic avian influenza (HPAI). In 2022, 67 countries across five continents reported H5N1 HPAI outbreaks in poultry and wild birds to the World Organisation for Animal Health (WOAH), resulting in over 131 million domestic poultry deaths or cullings (*WOAH, 2023*). In 2023, another 14 countries, mostly in the Americas, reported outbreaks. Several mass death events in wild birds were caused by influenza A(H5N1) clade 2.3.4.4b viruses (*Klaassen and Wille, 2023*; *WOAH, 2023*).

In that epidemiological context, understanding the environmental factors associated with the risk of HPAI circulation remains crucial. Extensive exploration of risk factors for HPAI presence, spread, and persistence has been conducted in various countries and regions. These studies have identified domestic waterfowl, several anthropogenic variables (human population density, distance to roads), and indicators of water presence as important factors for risk of H5N1 local circulation (*Gilbert and Pfeiffer, 2012*). More recent work on global suitability for HPAI has pinpointed that host-related variables such as poultry density are the strongest contributors to HPAI persistence (*Dhingra et al., 2016*; *Gierak and Śmietanka, 2021*; *Martin et al., 2011*).

In addition, poultry intensification, international poultry trade, live bird markets, and wild bird migratory routes have been recognised as playing a key role in the transmission and spread of HPAI (*Vandegrift et al., 2010*). Lastly, evidence of climate change impacting the dynamics of HPAI has also been discussed (*Prosser et al., 2023*). For example, changes in rainfall alter the distribution, abundance, and quality of wetlands, which can impact waterfowl populations (*Forcey et al., 2007*; *Gilbert et al., 2008*; *Vandegrift et al., 2010*). However, recent outbreaks raise questions about whether earlier ecological niche models still accurately predict the current distribution of areas ecologically suitable for the local circulation of HPAI H5 viruses. Ecological niche model outputs for range-shifting pathogens must therefore be interpreted with caution (*Elith et al., 2010*). Despite this limitation, correlative ecological niche models remain useful for identifying broad-scale associations and potential shifts in distribution. To account for this, we analysed two distinct time periods (2015–2020 and 2020–2023).

Understanding the interplay between environmental variables and avian influenza (AI) becomes increasingly crucial to have an in-depth understanding of the risk factors that govern avian influenza viruses (AIVs) circulation and spread. Following a previous study dedicated to the ecological niche of H5Nx and H5N1 (*Dhingra et al., 2016*), we aimed to address the following questions: (i) Which factors are associated with the increase in H5Nx and H5N1 cases since 2020? (ii) Do we observe a change or an extension of the ecological conditions suitable for local HPAI circulation? (iii) Are we able to predict the areas at risk for local HPAI circulation based on ecological niche models trained on occurrence data before 2020? To this end, we used HPAI H5N1 and H5Nx occurrence data collected before 2020 to train ecological niche models and tested whether these models could accurately predict areas ecologically suitable for HPAI circulation post-2020. We developed separate ecological niche models for wild and domestic bird HPAI occurrences, as these two groups are influenced by different ecological processes, surveillance biases, and management contexts. These models thus predict the ecological suitability for the risk of local viral circulation leading to the detection of HPAI occurrences within each host group (rather than the niche of the virus or the host groups alone). Specifically, we employed a boosted regression trees (BRT) approach to estimate the HPAI ecological suitability given local environmental conditions. We also computed the diversity indices of wild bird species involved in HPAI occurrences, helping to understand how shifts in species diversity might correspond

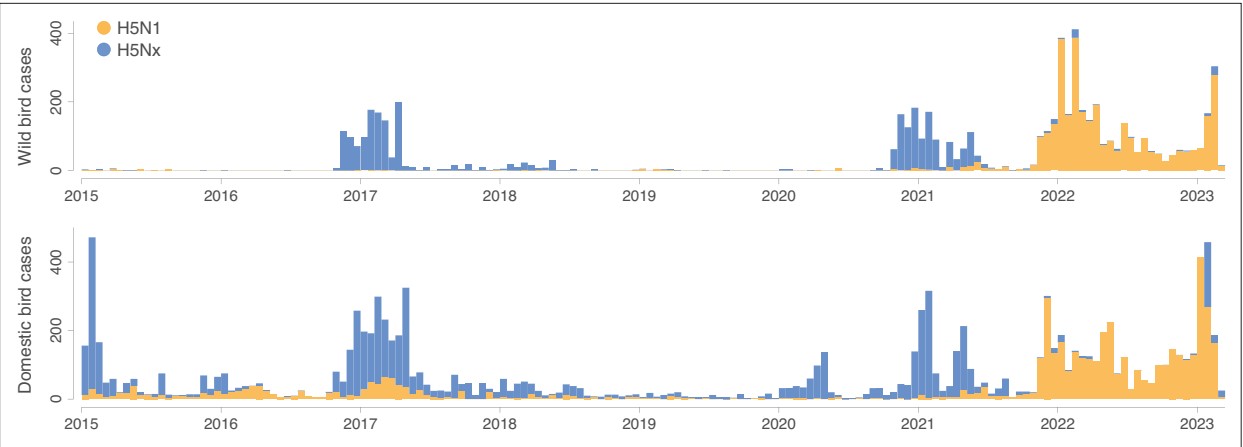

**Figure 1.** Epidemic curves for both wild and domestic bird cases. Orange and blue histograms report the weekly number of H5N1 cases and the weekly number of H5Nx cases, respectively.

## Results

In this study, we applied an ecological niche modelling approach using the BRT method to evaluate the distribution of areas ecologically suitable for the risk of local circulation of HPAI H5Nx and H5N1 viruses in wild and domestic bird populations across two periods: 2015–2020 and 2020–2022. We trained separate ecological niche models for each combination of host groups (wild or domestic birds), virus strain (H5Nx or H5N1), and time period. In addition to evaluating each model separately, models trained on data from the 2015 to 2020 period were tested to predict occurrences in the 2020–2022 period, allowing us to evaluate the potential of pre-2020 models to predict a more recent distribution of H5N1 and H5Nx cases.

We assessed the relative influence (RI) of a broad range of spatial predictors (*Elith et al., 2008*), including anthropogenic, topographical, land cover, eco-climatic, and poultry density, on virus occurrences (*Figure 2—figure supplement 1*). We used occurrence records from the EMPRES-i database and sampled pseudo-absence points to account for the lack of virus detection data in some areas. These pseudo-absence points were distributed based on human population density, with more pseudo-absence points sampled in areas of higher population to reflect the greater surveillance efforts in those regions (*Figure 2—figure supplements 3*).

The accuracy of the models was validated through three types of cross-validation: (i) a standard cross-validation with a random and stratified divide between training and validation sets, (ii) a spatial cross-validation based on the approach used by *Dhingra et al., 2016*, that clustered presence and pseudo-absence points into folds using reference presence points, and (iii) a second spatial cross-validation based on a blocks generation technique (*Valavi et al., 2019*). Models using spatial cross-validation techniques are less affected by spatial autocorrelation, as shown by higher spatial sorting bias (SSB) values (*Figure 2—figure supplement 4*). Among the seven datasets evaluated, four show better performance with the reference points-based spatial cross-validation method, suggesting it was less impacted by spatial autocorrelation compared to the block generation technique. Consequently, we have proceeded with the reference points approach for consistency with *Dhingra et al., 2016*.

Using pre-2020 data, our models demonstrated a relatively robust capability to predict the distribution of post-2020 occurrence data. The area under the curve (AUC) values for ecological niche

models trained on occurrence data in domestic birds before 2020 ranged between 0.74 and 0.77 when evaluated against post-2020 data, indicating a relatively good predictive performance. As expected, ecological niche models trained on post-2020 data showed a slightly higher predictive performance when evaluated on data from the same period, with AUC values ranging from 0.78 to 0.83 (*Supplementary file 1*). These results suggest that pre-2020 models captured broad patterns of suitability for H5Nx and H5N1 outbreaks, while post-2020 models provided a closer fit to the more recent epidemiological situation.

Previous literature reviews *Gilbert and Pfeiffer, 2012*; *Dhingra et al., 2016* have summarised the predictor variables commonly correlated with HPAI occurrences. As detailed in the Materials and methods section and in the table in Supplementary Information Resources S1, we explored four sets of environmental variables considered by *Dhingra et al., 2016*: host variables (set 1), land cover variables (set 2), eco-climatic variables (set 3), and a risk-based selection of variables performed by Dhingra and colleagues (set 4). Set 3 showed the lowest predictive performance for both domestic and wild bird cases during the two periods and, in contrast, sets 2 and 4 demonstrated the highest predictive accuracy for wild and domestic birds, respectively (*Figure 2—figure supplement 4*). Set 4 combined host variables with cultivated and managed vegetation, open water areas, distance to water, and the annual mean of land surface temperature. For domestic birds, set 1 (host variables) performed well before 2020, but its predictive performance decreased after 2020 in favour of set 4. Ecological niche models trained on wild and domestic bird cases, respectively, with the sets 2 and 4, are associated with AUC values exceeding 0.77, indicating a relatively strong performance in predicting H5 occurrences based on these sets of environmental variables.

*Figure 2* displays the response curves of the most important variables with their respective RI (*Supplementary file 1*). In the ecological niche models for H5N1 and H5Nx in domestic birds, the densities of intensive chicken populations, domestic duck populations, and human populations emerged as significant predictors, each with RI values exceeding 5%. Notably, the RI for intensive chicken density in H5N1 models increased sharply from 8.5% to 30.4% since 2020. Similarly, the RI of cultivated and managed vegetation has doubled for both strains post-2020. The response curves showed a positive correlation, indicating that higher values of these predictors were linked with an increased likelihood of HPAI occurrences given local environmental conditions. These findings indicate a trend towards increased HPAI susceptibility in environments characterised by intensive agricultural and vegetation management practices.

In contrast, eco-climatic factors such as land surface temperature and precipitation (set 3) showed only moderate influence. We also observed a decrease in the importance of domestic duck population density for both subtypes after 2020, possibly due to the increasing diversity of bird species involved in the transmission dynamics of H5N1 and H5NX (*Supplementary file 2*). These findings highlight the crucial role of anthropogenic and host-related variables in accurately predicting HPAI occurrences.

In the ecological niche models trained for wild bird cases, urban and built-up areas were associated with H5N1 and H5Nx occurrences (*Figure 2*), showing the highest RI prior to 2020. Specifically, RI values reached 54.5% before 2020 but decreased to 39.3% in the post-2020 period. This decline may indicate a reduced bias in observation data: typically, dead wild birds are more frequently found near human-populated areas due to opportunistic detections, whereas more recent surveillance efforts have become increasingly proactive (*Giacinti et al., 2024*). After 2020, we observed an increase in the importance of habitats, such as deciduous broadleaf trees (6.1%), mixed and other tree regions (11.4%), and herbaceous vegetation for H5Nx (9.5%). However, response curves (*Figure 2*) indicated that H5N1 and H5Nx occurrences in wild birds decreased in areas with greater tree cover, while they increased in regions dominated by herbaceous vegetation. Open water areas consistently showed high RI values across all time periods and virus strains, particularly for H5Nx before 2020 (25.5%) and H5N1 after 2020 (22.0%), highlighting their significant role in the ecological models of HPAI outbreaks.

*Figure 3* displays maps of the predicted ecological suitability for the risk of local H5N1 and H5NX circulation across two time periods, 2015–2020 and 2020–2022; these maps being also accessible on the following link for a dynamic visualisation of the results: https://app.mood-h2020.eu/core. All maps are in a WGS84 projection with a spatial resolution of 0.0833 decimal degrees (i.e. 5 arcmin, or approximately 10 km at the equator). For H5N1 and H5Nx in domestic birds, these maps reveal several regions associated with a relatively high ecological suitability, especially in Europe and Asia, in countries such as South Korea, Japan, Singapore, Malaysia, Vietnam, Cambodia, Thailand, and the

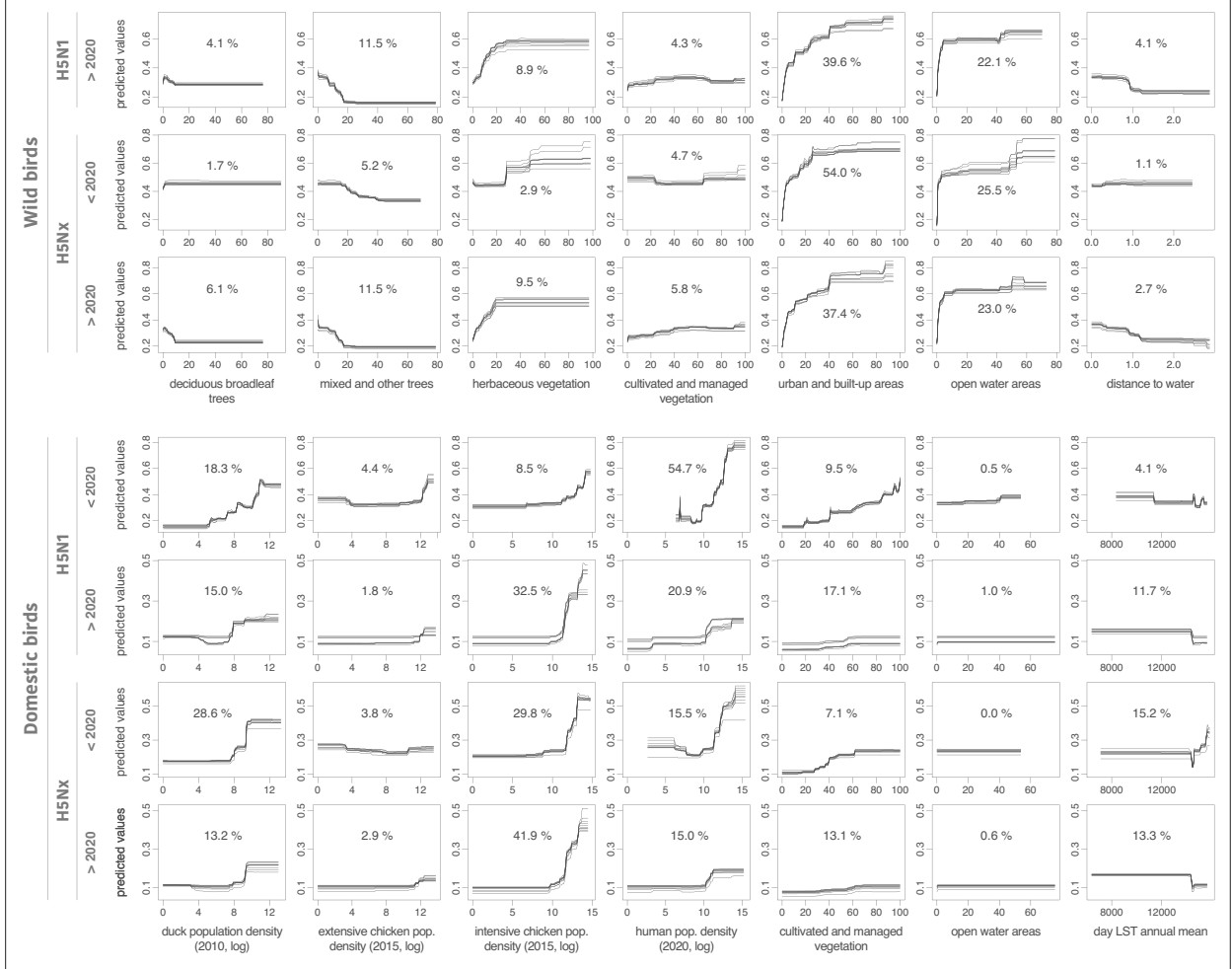

**Figure 2.** Response curves associated with the environmental variables included in the ecological niche models. For the ecological niche models trained on wild bird infection records, we here only display the response curves estimated for the environmental variables associated with an averaged relative influence (RI) >4% for at least one of the considered occurrence datasets (thus not reporting the response curves obtained for the following variables: evergreen deciduous needleleaf trees, evergreen broadleaf trees, shrublands, and regularly flooded vegetation). Each curve was retrieved from a distinct boosted regression tree (BRT) model trained for a specific dataset of occurrence data. We also report the averaged RI (in %) of each environmental variable in the respective ecological models trained on a specific dataset of occurrence data (see *Supplementary file 1* for the complete list of RI estimates along with their first and third quartiles). Due to a lack of data, the model was not trained for H5N1 in wild birds before 2020.

The online version of this article includes the following figure supplement(s) for figure 2:

**Figure supplement 1.** Environmental factors included in the ecological niche modelling.

**Figure supplement 2.** Comparison between the presence/pseudo-absence data used to train the ecological niche models (1° column) and the resulting ecological niche suitability maps (2° column).

**Figure supplement 3.** Comparison between the presence/pseudo-absence data used to train the ecological niche models (1° column) and the resulting ecological niche suitability maps (2° column).

**Figure supplement 4.** Boxplots reporting the spatial sorting bias (SSB) and area under the curve (AUC) of the receiving operator curve metrics associated with various ecological niche models trained for H5N1 and H5Nx.

Philippines, as well as the United Kingdom, France, the Netherlands, Germany, Italy, Ukraine, and Poland. Additionally, several areas in African nations, including Nigeria and South Africa, were identified as suitable environments for both H5N1 and H5Nx local circulation in scenarios before and after 2020. Regions in North America and South America – including Bolivia, Brazil, Colombia, Ecuador, Chile, Peru, and Venezuela – demonstrated ecological suitability for the risk of local circulation for H5Nx and H5N1, with a marked increase in the predicted suitability for H5N1 after 2020. For H5Nx post-2020, areas of high predicted ecological suitability, such as Brazil, Bolivia, the Caribbean islands,

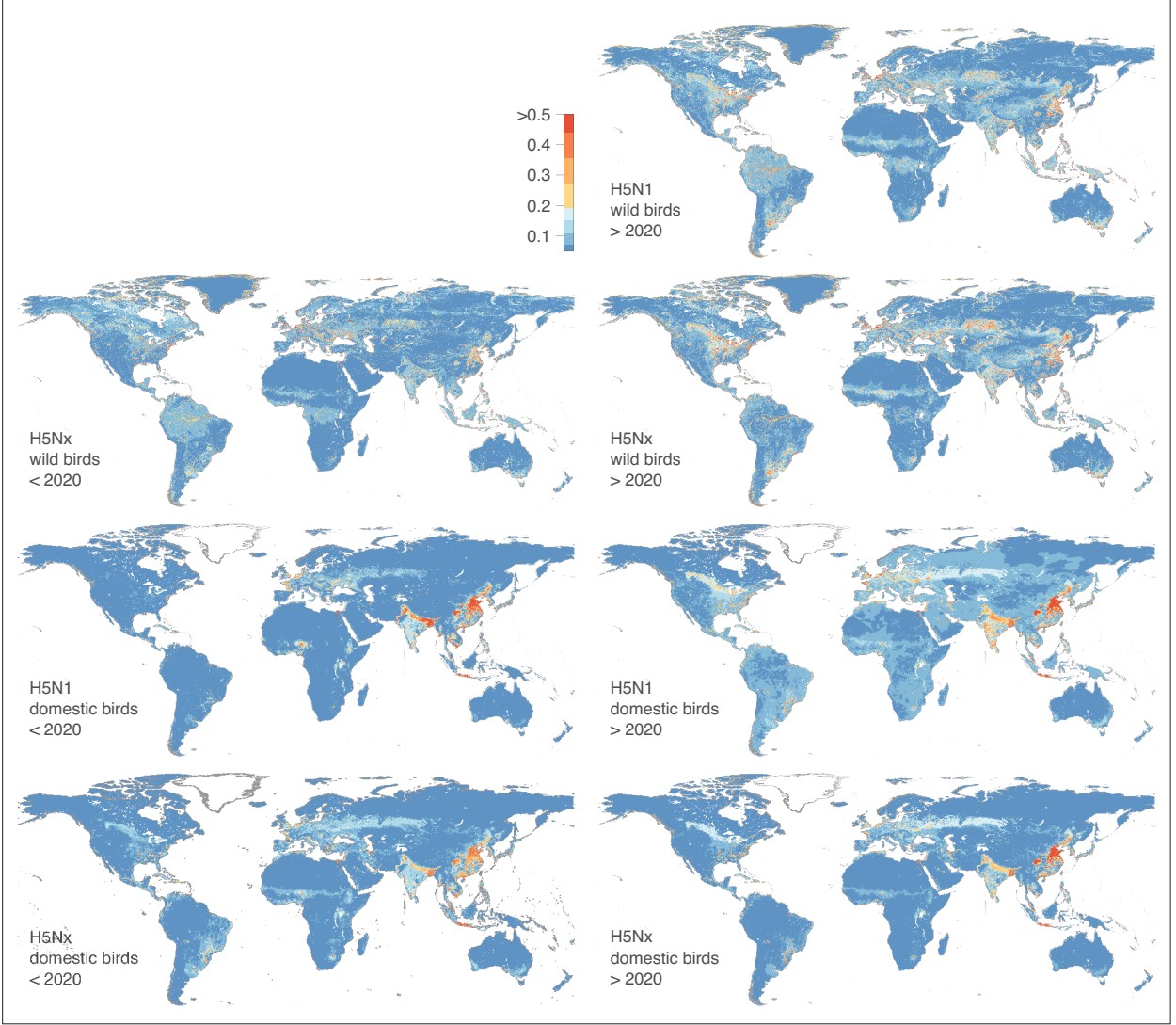

**Figure 3.** Areas ecologically suitable for local H5N1 or H5Nx circulation leading to infection cases in domestic bird populations. We estimated the ecological suitability for two different time periods (2015–2020 and 2020–2022) and for both wild and domestic bird populations. Dynamic visualisations of the results are available here: https://mood-platform.avia-gis.com/core.

The online version of this article includes the following figure supplement(s) for figure 3:

**Figure supplement 1.** Comparison between the ecological niche suitability estimated for H5N1 and H5Nx before 2016 and after 2020 for domestic bird populations.

and Jilin province in China, likely result from spatial extrapolations, as these regions reported few or no outbreaks in the training data.

Our findings were overall consistent with those previously reported by *Dhingra et al., 2016*, who used data from January 2004 to March 2015 for domestic poultry. However, some differences were noted: their maps identified higher ecological suitability for H5 occurrences before 2016 in North America, West Africa, eastern Europe, and Bangladesh, while our maps mainly highlight ecologically suitable regions in China, South-East Asia, and Europe (*Figure 3—figure supplement 1*). In India, analyses consistently identified high ecologically suitable areas for the risk of local H5Nx and H5N1 circulation for the three time periods (pre-2016, 2016–2020, and post-2020). Similar to the results reported by Dhingra and colleagues, we observed an increase in the ecological suitability estimated for H5N1 occurrence in South America's domestic bird populations post-2020. Finally, Dhingra and colleagues identified high suitability areas for H5Nx occurrence in North America, which are predicted to be associated with a low ecological suitability in the 2016–2020 models.

**Table 1.** Species diversity indices estimated from infected bird cases before and after 2020.

This table presents the Shannon and Simpson species diversity indices for various bird groups, comparing values before and after the year 2020. The indices are provided for all birds, sea birds, wild birds, domestic birds, as well as all birds affected by H5N1 and non-H5N1 strains.

| | All birds | | Sea birds | | Wild birds | | Domestic birds | | All birds H5N1 | | All birds non-H5N1 | |
|---|---|---|---|---|---|---|---|---|---|---|---|---|
| | *Shannon* | *Simpson* | *Shannon* | *Simpson* | *Shannon* | *Simpson* | *Shannon* | *Simpson* | *Shannon* | *Simpson* | *Shannon* | *Simpson* |
| <2020 | 4.227 | 0.971 | 2.187 | 0.843 | 3.072 | 0.832 | 3.919 | 0.966 | 2.003 | 0.833 | 3.415 | 0.945 |
| >2020 | 5.033 | 0.982 | 2.255 | 0.831 | 5.066 | 0.983 | 2.967 | 0.897 | 2.952 | 0.907 | 3.254 | 0.914 |

For H5Nx and H5N1 cases in wild birds, the estimated ecological niches were primarily associated with environmental factors such as open water and distance to water, which result in ecological suitability hotspots estimated near coasts and main rivers. For both H5Nx and H5N1, certain areas of predicted high ecological suitability appear spatially isolated, i.e., surrounded by regions of low predicted ecological suitability. These areas likely meet the environmental conditions associated with past HPAI occurrences, but their spatial isolation may imply a lower risk of actual occurrences, particularly in the absence of nearby outbreaks or relevant wild bird movements. Some of the areas with high predicted ecological suitability reflect the result of extrapolations. This is particularly the case in coastal regions of West and North Africa, the Nile Basin, Central Asia (Kyrgyzstan, Tajikistan, Uzbekistan), Brazil (including the Amazon and coastal areas), southern Australia, and the Caribbean, where ecological conditions are similar to those in areas where outbreaks are known to occur but where records of outbreaks are still rare.

We observed that areas of relatively high ecological suitability have expanded after 2020. North America, particularly near the Great Lakes region, appeared more suitable for local H5Nx circulation. This expansion was also evident in Russia and South America, aligning with major bird migration routes. Notably, while H5Nx had not been reported in Australia, the models indicate potential ecological suitability there as well. Expanded ecological suitability was also observed in China, India, and European countries. However, models trained on pre-2020 data maintained reasonable predictive performance when tested on post-2020 data, suggesting that the overall ecological niche of HPAI did not drastically shift over time.

In *Table 1*, we report the estimation of avian species diversity indices for species involved in HPAI outbreaks for the pre-2020 and post-2020 periods. Note that these indices reflect the diversity of bird species detected in outbreak records, not necessarily their abundance in the wild. We observed variations between these two periods both in the overall bird population and within specific wild bird species groups, including sea bird species (e.g. gulls, terns, boobies, gannets). For all birds, the Shannon index increased from 4.23 before 2020 to 5.03 after 2020, which might indicate a more diverse infected bird population in the latter period. The Simpson index, however, only rose from 0.97 to 0.98, but which could potentially suggest a slightly higher concentration of certain species post-2020. Sea birds exhibited a similar upward trend in diversity post-2020, with the Shannon index increasing from 2.19 to 2.26. This latter trend could potentially suggest a broader diversity of seabird species involved in HPAI outbreaks.

When considering wild birds affected by the H5N1 strain of HPAI, there is a notable increase in the Shannon index from 2.00 before 2020 to 2.95 after 2020, accompanied by a rise in the Simpson index from 0.83 to 0.91. These increases could be indicative of greater diversity and a more even distribution of wild bird species among the reported HPAI H5N1 wild bird cases after 2020. In contrast, birds not affected by the H5N1 strain showed a slight decrease in diversity post-2020. The Shannon index decreases from 3.42 to 3.26, and the Simpson index decreases from 0.95 to 0.92. Given that HPAI particularly affected sea birds >2020, we further explore the yearly distribution of sea bird families. As depicted in *Supplementary file 2*, positive cases among sea birds belonging to the families Laridae (gulls and terns), Sulidae (gannets and boobies), Ardeidae (herons), Pelecanidae (pelicans), Spheniscidae (penguins), Alcidae (auks), and Phalacrocoracidae (cormorants) were detected more frequently from 2020 in comparison to before 2020. Notably, the occurrence of Laridae and Sulidae species among positive cases increased markedly from 2021 onwards.

To evaluate whether the post-2020 increase in species diversity estimated for infected wild birds could result from an increase in the number of tests performed on wild birds, we compared European

annual surveillance test counts (*Aznar et al., 2025*; *Brouwer et al., 2019*) before and after 2020 using a Wilcoxon rank-sum test. We relied on European data because it was readily accessible and offered standardised and systematically collected metrics across multiple years, making it suitable for a comparative analysis. Although borderline significant (p-value=0.063), the Wilcoxon rank-sum test indeed highlighted a recent increase in the number of wild bird tests (on average >11,000/year pre-2020 and >22,000 post-2020), which indicates that the comparison of bird species diversity metrics should be interpreted with caution. However, such an increase in the number of tests conducted in the context of a passive surveillance framework would thus also be in line with an increase in the number of wild birds found dead and thus tested. Therefore, while the increase in the number of tests could indeed impact species diversity metrics such as the Shannon index, it can also reflect an absolute higher wild bird mortality in line with a broadened range of infected bird species.

## Discussion

Using an ecological niche modelling approach, our study presents an analysis of the ecological suitability, interpreted here as the risk of local circulation, for H5Nx and H5N1 HPAI viruses across different temporal and ecological contexts. By conducting BRT analyses and incorporating a diverse set of spatial predictors, our study provides insights into the environmental and anthropogenic factors that are correlated with the distribution of these viruses in both wild and domestic bird populations. For domestic birds, our results concur with those previously reported by *Dhingra et al., 2016*, indicating that ecological niche models incorporating host variables (chicken and duck population densities) outperform those based on land-use or eco-climatic factors in terms of predictive performance. Contrariwise, in our analysis of H5N1 and H5Nx HPAI in wild birds, the most significant predictors were anthropogenic factors, notably urban areas and distance to open water. However, observation biases might affect these findings since outbreaks in wild birds are commonly reported near human populations.

In our analyses, intensive chicken population density emerges as a significant predictor, underscoring the strong association of poultry farming practices with HPAI spread (*Dhingra et al., 2016*) and revealing a complex interplay between ecological niches and the anthropogenic landscape. We note, however, that intensive poultry density may reflect both surveillance intensity and epidemiological risk, and its predictive role in our models should be interpreted in light of both processes. Similarly to the results of *Dhingra et al., 2016*, land-use and climatic predictors do not play an important role in the niche ecological models, even for wild birds. This suggests that HPAI H5Nx is not as strongly environmentally constrained as vector-borne pathogens, for which clear eco-climatic boundaries (e.g. vector-borne diseases) can be mapped (*Dhingra et al., 2016*). Additionally, we found that while domestic duck population density is a significant predictor for HPAI occurrence in domestic birds before 2020, its importance has decreased post-2020, particularly within the H5Nx ecological niche model. This observation is particularly noteworthy, as domestic ducks have historically been closely linked to areas of persistence and evolution of H5N1 HPAI (*Gilbert and Pfeiffer, 2012*). However, our findings, in line with earlier studies such as the one of *Dhingra et al., 2016*, highlight a significant shift in the epidemiological dynamics of HPAI. Specifically, while the association between the H5Nx clade 2.3.4.4 and domestic ducks has, for instance, been documented in South Korea (*Hill et al., 2015*), it is becoming increasingly evident that HPAI is now more strongly associated with high-density chicken farming and anthropogenic factors. This shift suggests that the virus could now be circulating more among farms (farm-to-farm transmission) (*FAO, 2023*), with fewer introductions by wild birds – particularly wild ducks, which have previously acted as 'Trojan horses' for H5N1 HPAI (*Kim et al., 2014*; *Park and Kim, 2017*). However, this remains an interpretation, as the available data do not allow us to distinguish between index cases and secondary transmission events. Consequently, we could be witnessing a shift from the traditional duck/rice ecosystems towards a scenario where intensive chicken farming and a wider diversity of wild birds play a more prominent role in the transmission dynamics of HPAI. Therefore, monitoring areas with high intensive chicken densities and regular surveillance of wild birds remains crucial for the early detection and management of HPAI outbreaks.

A difference between the ecological niche models previously trained by *Dhingra et al., 2016* and our models is the pronounced increase in ecological suitability for H5N1 in domestic bird populations estimated in India and China. More generally, our models highlight the extensive spread of H5N1 and H5Nx HPAI among domestic birds in Asia previously identified as 'reassortment sink areas' (*Dhingra*

*et al., 2018*). Reassortment refers to the process where influenza viruses exchange genetic material, potentially leading to new, more virulent strains. The extent of virus suitability in Asia highlights a potential change in the dynamics of virus spread, possibly driven by intensified poultry farming practices and increased human-wildlife interactions in these regions (*Bahl et al., 2016*; *Gilbert et al., 2017*).

The recent outbreaks of infection in cattle within the United States have been partially delineated by our HPAI risk mapping approach, highlighting major hotspots predominantly in the Great Lakes region, including Michigan. From March to July 2024, approximately 26 herds in Michigan were affected (*U.S. Department of Agriculture A and PHIS, 2026*). Additionally, our risk maps identify localised hotspots around urban centres in Colorado and Texas, where herds have also been infected. These outbreaks appear to stem from a single introduction event in Texas by wild birds (*Worobey et al., 2024a*; *Worobey et al., 2024b*). This trend may be indicative of an increasing risk of future introductions from wild birds, as we observed a significant rise in the diversity of infected bird species after 2020, particularly among sea birds.

The latest strain of H5N1 (H5 clade 2.3.4.4b), believed to have emerged in 2014, shows adaptations for infecting a wide range of wild birds and mammals (*Caliendo et al., 2022*; *Graziosi et al., 2024*). By 2020, its spread among wild birds was three times faster than in farmed poultry, facilitated by extensive reassortment with low-pathogenic avian influenza viruses and mutations associated with adaptation to wild bird hosts (CDC, 2024 *Xie et al., 2023*;). Unlike previous epidemic waves in Europe, outbreaks in the last years have also continued through summer, confirming that the virus's current circulation is no longer seasonal but nesting in the wild bird population (*Sciensano, 2024*). Therefore, the virus is able to infect a wider range of species than previous forms, as it circulates all seasons (*Sacristán et al., 2024*). The densely packed colonies of sea birds have also facilitated the rapid virus spread, resulting in high mortality in these sea bird colonies (*Boulinier, 2023*; *Bregnballe et al., 2024*). For instance, the Netherlands lost up to 80% of its Sandwich terns in weeks, and the United Kingdom saw significant outbreaks, especially on Scottish islands (*Knief et al., 2024*). In South America, similar trends have been observed, with notable die-offs in seabird populations along coastal regions, particularly impacting species such as penguins, pelicans, and cormorants (*Leguia et al., 2023*). Since then, in early 2023, the Peruvian outbreak had spread to marine mammals, particularly affecting the South American sea lion, which also began to experience a mass die-off (*Campagna et al., 2024*; *Leguia et al., 2023*).

The increase in the number of wild bird species being infected by HPAI, particularly H5N1, could be attributed to several factors such as viral evolution, increased interaction between domestic and wild birds, climate and land-use changes, among others. Regarding land-use changes, large parts of Europe have become increasingly fragmented due to the expansion of urban and transport infrastructure (*European Environment Agency, 2022*). This habitat loss and fragmentation can significantly impact water availability and trophic resources (*Santos-Tovar et al., 2024*), which could in turn aid the spread and transmission of AIVs due to relocation of birds and outbreaks (*Yin et al., 2022*). Climate change has also been reported as a factor that could affect wild bird distribution in different ways (*Gilbert et al., 2008*). As global climate conditions change, avian migratory patterns and routes are also changing (*Hitch and Leberg, 2007*; *van Doren, 2022*). Furthermore, higher temperatures and extreme weather have resulted in large-scale population shifts in a range of temperate species (*Dezfuli et al., 2022*). Overall, these elements therefore raise the question of the role of climate and land-use changes in the recent shifting epidemiology of HPAI in wild birds.

The increasing diversity of infected bird species is part of a broader resurgence of H5N1 globally. Phylogenetic analysis from Europe revealed significant genetic diversity in HPAI-H5N1 viruses since October 2020, likely due to multiple reassortment events with LPAI viruses. Notable introductions from Russia to Europe were observed through autumn migrations of wild birds (*ECDC, 2021*). Additionally, the transatlantic transport of HPAI H5N1, documented in Canada in December 2021, suggests the virus can spread across continents, a route previously seen only with LPAI viruses (*Caliendo et al., 2022*; *Dusek et al., 2014*; *Huang et al., 2014*). In November 2022, H5N1 (clade 2.3.4.4b) was introduced into South America, presumably through the movements of migratory wild birds travelling south during the boreal winter. The virus spread rapidly and caused significant mortality in wild birds and marine mammals across multiple countries (*Leguia et al., 2023*; *Banyard et al., 2023*; *Campagna et al., 2024*; *Gamarra-Toledo et al., 2023*; *WOAH, 2024*).

Interestingly, our models spatially extrapolate ecological suitability for H5Nx in wild birds in areas where few or no outbreaks have been reported. This discrepancy may be explained by limited surveillance or underreporting in those regions. For instance, there is significant evidence that Kazakhstan and Central Asia play a role as a centre for the transmission of AIVs through migratory birds (*Amirgazin et al., 2022*; *FAO, 2005*; *Sultankulova et al., 2024*). However, very few wild bird cases are reported in EMPRES-i. In contrast, Australia appears environmentally suitable in our models, yet no incursion of HPAI H5N1 2.3.4.4b has occurred despite the arrival of millions of migratory shorebirds and seabirds from Asia and North America. Extensive surveillance in 2022 and 2023 found no active infections nor evidence of prior exposure to the 2.3.4.4b lineage (*Wille et al., 2024*; *Wille and Klaassen, 2023*).

Despite the robustness of our ecological niche models, some limitations must be acknowledged. Firstly, the accuracy of our models is inherently dependent on the quality and completeness of the occurrence data used, which is influenced by the different reporting systems in place in different countries. Secondly, the definitions of key explanatory variables, such as intensive versus extensive production, may vary across regions, adding complexity to global predictions. To address these limitations, future work could focus on developing country-level or regional models that incorporate localised data and context-specific definitions. Thirdly, underreporting and biases in outbreak reports, particularly in wild bird populations, can skew model predictions and underestimate areas at risk. Furthermore, another limitation of our models for wild bird populations is the lack of incorporation of migratory patterns and behaviours. Migratory birds have complex movement patterns influenced by seasonal changes, food availability, and habitat conditions. Our models, using static spatial predictors, do not capture these dynamic routes and stopover sites, leading to potential inaccuracies in predicting areas at risk for HPAI circulation. In future work, the integration of dynamic migratory data and environmental changes into our ecological niche models could provide a more comprehensive and accurate risk assessment for HPAI spread among wild bird populations. In addition, aligning outbreak occurrences with seasonally matched environmental variables could further refine predictions of HPAI risk linked to migratory dynamics. Finally, our models for domestic poultry do not distinguish between primary introduction events (e.g. spillover from wild birds) and secondary transmission between farms due to limitations in the available surveillance data. While environmental factors likely influence the risk of initial spillover events, secondary spread is more often driven by anthropogenic factors such as biosecurity practices and poultry trade, which are not included in our current modelling framework.

Despite these limitations, our study provides significant contributions to the understanding of H5N1 and H5Nx HPAI virus dynamics. By integrating a wide range of spatial predictors and using ecological niche modelling techniques, we provide a quantitative analysis on the factors driving virus distribution. Overall, our study underscores the importance of continuous surveillance and the need for adaptive management strategies to address the evolving threat of HPAI viruses globally.

## Materials and methods

Our approach is similar to the one implemented by *Dhingra et al., 2016*. While *Dhingra et al., 2016*, developed their models only for domestic birds over the 2003–2016 period, our models were developed for two host groups separately (wild and domestic birds) and for two time periods (2016–2020 and 2020–2023).

### HPAI occurrence data acquisition

The dataset containing HPAI occurrences was constructed by extracting relevant information from the EMPRES-i database spanning the period from January 5, 2015, to March 7, 2023. This dataset encompasses comprehensive details pertaining to H5N1 and H5Nx subtypes, including the family of birds involved in transmission, observation and reporting dates, location coordinates, and the respective countries where the reports originated. The dataset comprises a total of 17,410 H5Nx and 8,383 H5N1 reported cases. It is important to note that the EMPRES-i database does not distinguish between index cases (e.g. primary spillover from wild birds) and secondary farm-to-farm transmissions. As such, our ecological niche models are trained on confirmed HPAI outbreaks in poultry that may result from different transmission dynamics, including both initial introduction events influenced by environmental factors and subsequent spread within poultry systems.

## Environmental data acquisition

Previous literature reviews *Gilbert and Pfeiffer, 2012*; *Dhingra et al., 2016* have summarised the predictor variables commonly linked to occurrences of HPAI. Here, we explored four sets of environmental variables considered by *Dhingra et al., 2016*, and which are detailed in the table in Supplementary Information Resources S1: host variables (set 1), land cover variables (set 2), eco-climatic variables (set 3), and a risk-based selection of variables performed by Dhingra and colleagues (set 4).

Set 1 includes $\log_{10}$-transformed extensive and intensive chicken density (*Gilbert et al., 2015*), domestic duck density (*Robinson et al., 2014*), and human population density from the Gridded Population of the World (GPW) (*Center For International Earth Science Information Network-CIESIN-Columbia University, 2017*) database. The chicken density data includes GIS layers representing the global distribution of chickens, categorised into extensive and intensive systems. It uses the model published by *Gilbert et al., 2018*, applied to the Gridded Livestock of the World (GLW) version 3 chicken layers (*Gilbert et al., 2022b*). The global domestic duck distribution data were computed using the GLW version 4 (*Gilbert et al., 2022a*). Set 2 includes land cover information, which was obtained from the global 1 km Consensus Land Cover database by *Tuanmu and Jetz, 2014*, and identifies different land cover categories by providing a percentage of representation of each land cover category within raster cells of approximately 1 km$^2$ (http://www.earthenv.org/landcover.html). Additionally, this dataset was supplemented by a layer about the distance of each spatial point to the open water. Set 3 encompasses seasonal and large-scale patterns of eco-climatic indices like daytime land surface temperature and the normalised difference vegetation index, as documented by *Scharlemann et al., 2008*. Finally, set 4 is a selection from the previous sets based on prior epidemiological knowledge (*Gilbert and Pfeiffer, 2012*; *Dhingra et al., 2016*).

## Ecological niche modelling

The different ecological niche modelling analyses were performed with the BRT, available in the R package 'dismo' (*Hijmans et al., 2017*). BRT is a machine learning approach that can be used to generate a collection of sequentially fitted regression trees optimising the predictive probability of occurrence given local environmental conditions (*Elith et al., 2006*). The predictive probability, interpreted as ecological suitability for local virus circulation, ranges from '0' (unsuitable environmental conditions) to '1' (highly suitable environmental conditions). BRT allows us to model complex non-linear relationships between the response and various predictor variables (*Elith et al., 2008*). It has also been demonstrated that the BRT approach had a superior predictive performance compared to alternative modelling approaches (*Elith et al., 2006*).

We selected 2020 as a cut-off point to reflect a well-documented shift in HPAI epidemiology, notably the emergence and global spread of clade 2.3.4.4b. This event marked a turning point in viral dynamics, influencing both the range of susceptible hosts and the geographical distribution of outbreaks. Based on this distinction, we performed ten replicated BRT analyses using each of the seven occurrence datasets: (i) H5Nx cases in wild birds <2020 (01/01/2015 to 31/12/2019), (ii) H5N1 cases in wild birds >2020 (01/01/2020 to 06/03/2023), (iii) H5Nx cases in wild birds >2020 (01/01/2020 to 06/03/2023), (iv) H5Nx cases in domestic birds <2020 (01/01/2015 to 31/12/2019), (v) H5N1 cases in domestic birds <2020 (01/01/2015 to 31/12/2019), (vi) H5N1 cases in domestic birds >2020 (01/01/2020 to 06/03/2023), and (vii) H5Nx cases in domestic birds >2020 (01/01/2020 to 06/03/2023). Of note, there was not enough occurrence data to train ecological niche models on H5N1 cases in wild birds <2020 (only 33 occurrence data from eight different countries – Bangladesh, Bulgaria, China, India, Kazakhstan, Nepal, Romania, and Russia – between the 01/01/2015 and 31/12/2019).

We ran BRT analyses using both occurrence and 'pseudo-absence' data, the latter corresponding to randomly sampled absence points within the study area (*Elith et al., 2006*). To account for heterogeneity in AIV surveillance and minimise the risk of sampling pseudo-absences in poorly monitored regions, we restricted our analysis to countries (or administrative level 1 units in China and Russia) with at least five confirmed outbreaks. Unlike *Dhingra et al., 2016*, who sampled pseudo-absences across a broader global extent, our sampling was limited to regions with demonstrated surveillance activity. In addition, we adjusted the density of pseudo-absence points according to the number of reported outbreaks in each country or admin-1 unit, as a proxy for surveillance effort – an approach not implemented in this previous study.

In addition, the sampling pseudo-absence points were based on the human population distribution to further account for potential within-country differences in surveillance intensity (*Dhingra et al., 2016*): the probability to sample a pseudo-absence from a given grid cell was proportional to the log-transformed human population density in that cell. The human population density raster was retrieved from the GPW project, which was downscaled to a resolution of 5 arcminutes. Another approach to sampling pseudo-absences would have been to distribute them according to the density of domestic poultry. However, as pointed out by *Dhingra et al., 2016*, the locations of outbreaks in the EMPRES-i database are often georeferenced using place name nomenclatures due to a lack of accurate GPS data, which could introduce a spatial bias towards populated areas. Therefore, the use of human population density as an indicator of surveillance effort takes into account not only the probability of detection, but also the geo-referencing bias in household records. To further reduce bias from unsuitable regions where the pathogen may not have been introduced, we restricted pseudo-absence sampling to areas located at a minimum of 10 km and a maximum of 1000 km from known presence points (*Phillips et al., 2009*; *Dhingra et al., 2016*). Initially, 10 times more pseudo-absences than presence points were simulated, but we ultimately retained three times more pseudo-absences than presences in each country/admin-1 unit.

Given that spatial data typically exhibits autocorrelation, employing a standard cross-validation procedure may result in an overestimation of the BRT model's accuracy (*Randin et al., 2006*). To circumvent this issue, we employed a spatial cross-validation procedure to select the optimal number of trees in each BRT model. Specifically, we tested and compared two distinct spatial cross-validation procedures: (i) the procedure introduced by Dhingra and colleagues and based on (five) reference presence points used to cluster the presence and pseudo-absence points in (five) folds according to their nearest distance to a reference point, and (ii) the procedure introduced by *Valavi et al., 2019*, and based on a block generation method. In the first spatial cross-validation procedure, the five reference presence points were randomly selected among all presence points. If each of the five generated spatial folds did not gather at least 10% of the presence points, the random selection of reference points was re-initiated. To compare the performance of the three cross-validation procedures applied in this study (the standard procedure and the spatial procedures based on the reference points or blocks generation), we estimated the SSB metric (*Hijmans, 2012*) ranging from 0 to 1, an SSB near 0 and 1, indicating an extreme impact and the absence of an impact of spatial autocorrelation on the model training, respectively.

All BRT analyses were run and averaged over 10 cross-validated replicates, with a tree complexity of 4, a learning rate of 0.01, a tolerance parameter of 0.001, and while considering 5 spatial folds. Each model was initiated with 10 trees, and additional trees were incrementally added (in steps of 5) up to a maximum of 10,000, with the optimal number selected based on cross-validation tests. We evaluated the inferences using the area under the receiver operating characteristic curve, also simply referred to as AUC. We further used the AUC metric to assess the capacity of our models trained on occurrence data <2020 to predict >2020 distribution of occurrence data. To assess how each environmental factor contributed to the different BRT models, we computed their RI in those models and their response curves, showing how ecological suitability varies with one specific factor. RI values were obtained by evaluating the number of times a specific environmental factor was selected for splitting a tree, weighted by the squared improvement to the model as a result of each split, averaged over all trees (*Elith et al., 2008*).

We explored the four sets of environmental variables assembled by *Dhingra et al., 2016*, and described above in the 'Environmental data acquisition' subsection. The BRT analyses presented here were eventually performed based on set 2 for wild bird cases and set 4 for domestic bird cases. This selection is aligned with the methodology of *Dhingra et al., 2016*, prioritising variables which demonstrated the best predictive accuracy and clear epidemiological significance to minimise the potential for coincidental correlations. The RI values obtained when analysing each set of environmental variables separately are all reported in a table in Supplementary Information Resources S2.

## Bird diversity indices calculation

Diversity index calculations were performed using the HPAI dataset. Cases lacking information on the date of collection, bird family, and subtype were excluded from the analyses. The diversity of HPAI cases across bird families was quantified using two indices: the Shannon diversity index and the

Simpson diversity index. The Shannon index accounts for evenness of bird families (*Shannon, 1948*), whereas the Simpson index provides a measure of dominance (*Simpson, 1949*), emphasising the most abundant families in the dataset. Simpson index ranges between 0 and 1, where 1 represents infinite diversity and 0 no diversity. Calculations were done using two time points: before and after 2020, and for various data classifications: overall and annually, across all bird families and specifically sea bird families, as well as for all subtypes (HxNx) and those identified as H5N1 subtype, wild birds, and poultry. These indices were calculated using the R package 'vegan' (*Oksanen et al., 2013*): the dataset was first aggregated by year, HPAI subtype, and bird family, and we then used the 'diversity' function to compute the Shannon and Simpson indices for each year.

## Acknowledgements

MFVG and SD acknowledge funding from the European Union Horizon 2020 project MOOD (grant agreement n°874850). MCD was supported by the UKRI GCRF One Health Poultry Hub (Grant No. B/S011269/1), one of twelve interdisciplinary research hubs funded under the UK government's Global Challenge Research Fund Interdisciplinary Research Hub initiative. MCD and SD are supported by the BELSPO project BE-PIN. SD also acknowledges support from the *Fonds National de la Recherche Scientifique* (F.R.S.-FNRS, Belgium; grant n°F.4515.22), from the Research Foundation – Flanders (*Fonds voor Wetenschappelijk Onderzoek – Vlaanderen*, FWO, Belgium; grant n°G098321N), from the European Union Horizon 2020 project LEAPS (grant agreement n°101094685), from the Immun-Reach project funded by the *Institut d'Encouragement de la Recherche Scientifique et de l'Innovation de Bruxelles* (Innoviris, Belgium), and from the doctoral network VIVACE funded by the Marie Skłodowska-Curie Actions (MSCA) of the European Commission (grant agreement n°101167768).

## Additional information

### Competing interests

William Wint: employee of Environmental Research Group Oxford Ltd. Guy Hendrickx: Guy Hendrickx is affiliated with Avia-GIS. The author has no other competing interests to declare. Cedric Marsboom: Cedric Marsboom is affiliated with Avia-GIS. The author has no other competing interests to declare. The other authors declare that no competing interests exist.

### Funding

| Funder | Grant reference number | Author |
|---|---|---|
| Horizon 2020 Framework Programme | 874850 | Maria F Vincenti-Gonzalez Simon Dellicour |
| UK Research and Innovation | B/S011269/1 | Marie-Cécile Dupas |
| Fonds De La Recherche Scientifique - FNRS | F.4515.22 | Simon Dellicour |
| Belgian Federal Science Policy Office | TD/231/BE-PIN | Marie-Cécile Dupas Simon Dellicour |
| Fonds Wetenschappelijk Onderzoek | G098321N | Simon Dellicour |
| Horizon 2020 Framework Programme | 101094685 | Simon Dellicour |
| Marie Skłodowska-Curie Actions (MSCA) of the European Commission | 101167768 | Simon Dellicour |

The funders had no role in study design, data collection and interpretation, or the decision to submit the work for publication.

## Author contributions

Marie-Cécile Dupas, Conceptualization, Data curation, Validation, Investigation, Methodology, Writing – original draft, Writing – review and editing; Maria F Vincenti-Gonzalez, Conceptualization, Data curation, Validation, Methodology, Writing – original draft, Project administration, Writing – review and editing; Madhur Dhingra, Data curation, Writing – review and editing; Claire Guinat, Timothée Vergne, Writing – review and editing; William Wint, Funding acquisition, Writing – review and editing; Guy Hendrickx, Funding acquisition, Visualization, Project administration, Writing – review and editing; Cedric Marsboom, Visualization, Project administration, Writing – review and editing; Marius Gilbert, Conceptualization, Funding acquisition, Methodology, Writing – review and editing; Simon Dellicour, Conceptualization, Formal analysis, Supervision, Funding acquisition, Validation, Investigation, Methodology, Writing – original draft, Project administration, Writing – review and editing

## Author ORCIDs

Marie-Cécile Dupas  https://orcid.org/0000-0002-8769-2243
Simon Dellicour  https://orcid.org/0000-0001-9558-1052

Reviewer #1 (Public review): https://doi.org/10.7554/eLife.104748.4.sa1
Reviewer #2 (Public review): https://doi.org/10.7554/eLife.104748.4.sa2
Author response https://doi.org/10.7554/eLife.104748.4.sa3

# Additional files

## Supplementary files

Supplementary file 1. Supplementary tables reporting model performance metrics (area under the curve [AUC]) and relative influence estimates of environmental variables used in the ecological niche models.

Supplementary file 2. Distribution of H5Nx and H5N1 occurrence records in sea birds from 2015 to 2023, categorised by bird family. The three panels successively show the total occurrence records for all H5Nx subtypes, occurrence records for non-H5N1 H5Nx subtypes, and all H5N1 occurrence records.

MDAR checklist

## Data availability

All data were retrieved from EMPRES-i database, and are publicly available. R scripts and related files needed to run the analyses, as well as Supplementary Information Resources S1 (description and source of each environmental variable included in the original sets of variables) and S2 (mean relative influence computed for the environmental variables within each considered set of variables), are all available at https://doi.org/10.5281/zenodo.18325712.

The following previously published dataset was used:

| Author(s) | Year | Dataset title | Dataset URL | Database and Identifier |
| --- | --- | --- | --- | --- |
| Dellicour S, Dupas M-C | 2026 | sdellicour/h5nx_risk_mapping: release | https://doi.org/10.5281/zenodo.18325712 | Zenodo, 10.5281/zenodo.18325712 |

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
