## [Editor Report · eLife Assessment]

This global study compares environmental niche model outputs of avian influenza pathogen niche constructed for two distinct periods, and uses differences between those outputs to suggest that the changed case numbers and distribution relate to intensification of chicken and duck farming, and extensive cultivation. While a **useful** update to existing niche models of highly pathogenic avian influenza, the justification for the use of environmental niche models to explore land cover change as a driver of changed case epidemiology is **incomplete**.

---

## [Referee Report · Reviewer #1 (Public review)]

The authors aim to predict ecological suitability for transmission of highly pathogenic avian influenza (HPAI) using ecological niche models. This class of models identify correlations between the locations of species or disease detections and the environment. These correlations are then used to predict habitat suitability (in this work, ecological suitability for disease transmission) in locations where surveillance of the species or disease has not been conducted. The authors fit separate models for HPAI detections in wild birds and farmed birds, for two strains of HPAI (H5N1 and H5Nx) and for two time periods, pre- and post-2020. The authors also validate models fitted to disease occurrence data from pre-2020 using post-2020 occurrence data.

---

## [Referee Report · Reviewer #2 (Public review)]

Summary:

The geographic range of highly pathogenic avian influenza cases changed substantially around the period 2020, and there is much interest in understanding why. Since 2020 the pathogen irrupted in the Americas and the distribution in Asia changed dramatically. This study aimed to determine which spatial factors (environmental, agronomic and socio-economic) explain the change in numbers and locations of cases reported since 2020 (2020--2023). That's a causal question which they address by applying correlative environmental niche modelling (ENM) approach to the avian influenza case data before (2015--2020) and after 2020 (2020--2023) and separately for confirmed cases in wild and domestic birds. To address their questions they compare the outputs of the respective models, and those of the first global model of the HPAI niche published by Dhingra et al 2016.

ENM is a correlative approach useful for extrapolating understandings based on sparse geographically referenced observational data over un- or under-sampled areas with similar environmental characteristics in the form of a continuous map. In this case, because the selected covariates about land cover, use, population and environment are broadly available over the entire world, modelled associations between the response and those covariates can be projected (predicted) back to space in the form of a continuous map of the HPAI niche for the entire world.

Strengths:

The authors are clear about expected bias in the detection of cases, such geographic variation in surveillance effort (testing of symptomatic or dead wildlife, testing domestic flocks) and in general more detections near areas of higher human population density (because if a tree falls in a forest and there is no-one there, etc), and take steps to ameliorate those. The authors use boosted regression trees to implement the ENM, which typically feature among the best performing models for this application (also known as habitat suitability models). They ran replicate sets of the analysis for each of their model targets (wild/domestic x pathogen variant), which can help produce stable predictions. Their code and data is provided, though I did not verify that the work was reproducible.

The paper can be read as a partial update to the first global model of H5Nx transmission by Dhingra and others published in 2016 and explicitly follows many methodological elements. Because they use the same covariate sets as used by Dhingra et al 2016 (including the comparisons of the performance of the sets in spatial cross-validation) and for both time periods of interest in the current work, comparison of model outputs is possible. The authors further facilitate those comparisons with clear graphics and supplementary analyses and presentation. The models can also be explored interactively at a weblink provided in text, though it would be good to see the model training data there too.

The authors' comparison of ENM model outputs generated from the distinct HPAI case datasets is interesting and worthwhile, though for me, only as a response to differently framed research questions.

Weaknesses:

This well-presented and technically well-executed paper has one major weakness to my mind. I don't believe that ENM models were an appropriate tool to address their stated goal, which was to identify the factors that "explain" changing HPAI epidemiology.

Comments on the revised version from the editors:

We are extremely grateful to the authors for presenting a thoughtful and respectful point by point rebuttal to the prior reviewers' comments. After reading these comments carefully, we conclude that there is a straightforward strongly held disagreement between the authors and the reviewers as to the validity of the methods (Ecological Niche Modeling) for this particular dataset. Please note that the two reviewers have substantial expertise in the area of Ecologic Niche Modeling. We elected not to reach out to the reviewers for a third set of comments as we do not think their overall opinions will change, and wish to be respectful of their time.

To allow readers a balanced assessment of the paper, we intend to publish your rebuttal comments in full. It is our hope that interested readers can weigh both sides of this respectful and interesting debate in order to reach their own conclusions about the strength of evidence presented in your manuscript.

---

## [Author Response]

The following is the authors’ response to the previous reviews

**Public Reviews:**

We thank the Reviewers for their thorough attention to our paper and the interesting discussion about the findings. Before responding to more specific comments, here some general points we would like to clarify:

(1) Ecological niche models are indeed correlative models, and we used them to highlight environmental factors associated with HPAI outbreaks within two host groups. We will further revise the terminology that could still unintentionally suggest causal inference. The few remaining ambiguities were mainly in the Discussion section, where our intent was to interpret the results in light of the broader scientific literature. Particularly, we will change the following expressions:

- “Which factors can explain…” to “Which factors are associated with…” (line 75);

- “the environmental and anthropogenic factors influencing” to “the environmental and anthropogenic factors that are correlated with” (line 273);

- “underscoring the influence” to “underscoring the strong association” (line 282).

(2) We respectfully disagree with the suggestion that an ecological niche modelling (ENM) approach is not appropriate for this work and the research question addressed therein. Ecological niche models are specifically designed to estimate the spatial distribution of the environmental suitability of species and pathogens, making them well suited to our research questions. In our study, we have also explicitly detailed the known limitations of ecological niche models in the Discussion section, in line with prior literature, to ensure their appropriate interpretation in the context of HPAI.

(3) The environmental layers used in our models were restricted to those available at a global scale, as listed in Supplementary Information Resources S1 (https://github.com/sdellicour/h5nx\_risk\_mapping/blob/master/Scripts\_%26\_data/SI\_Resource\_S1.xlsx). Naturally, not all potentially relevant environmental factors could be included, but the selected layers are explicitly documented and only these were assessed for their importance. Despite this limitation, the performance metrics indicate that the models performed well, suggesting that the chosen covariates capture meaningful associations with HPAI occurrence at a global scale.

**Reviewer #1 (Public review):**
The authors aim to predict ecological suitability for transmission of highly pathogenic avian influenza (HPAI) using ecological niche models. This class of models identify correlations between the locations of species or disease detections and the environment. These correlations are then used to predict habitat suitability (in this work, ecological suitability for disease transmission) in locations where surveillance of the species or disease has not been conducted. The authors fit separate models for HPAI detections in wild birds and farmed birds, for two strains of HPAI (H5N1 and H5Nx) and for two time periods, pre- and post-2020. The authors also validate models fitted to disease occurrence data from pre-2020 using post-2020 occurrence data. I thank the authors for taking the time to respond to my initial review and I provide some follow-up below.Detailed comments:In my review, I asked the authors to clarify the meaning of "spillover" within the HPAI transmission cycle. This term is still not entirely clear: at lines 409-410, the authors use the term with reference to transmission between wild birds and farmed birds, as distinct to transmission between farmed birds. It is implied but not explicitly stated that "spillover" is relevant to the transmission cycle in farmed birds only. The sentence, "we developed separate ecological niche models for wild and domestic bird HPAI occurrences ..." could have been supported by a clear sentence describing the transmission cycle, to prime the reader for why two separate models were necessary.

We respectfully disagree that the term “spillover” is unclear in the manuscript. In both the Methods and Discussion sections (lines 387-391 and 409-414), we explicitly define “spillover” as the introduction of HPAI viruses from wild birds into domestic poultry, and we distinguish this from secondary farm-to-farm transmission. Our use of separate ecological niche models for wild and domestic outbreaks reflects not only the distinction between primary spillover and secondary transmission, but also the fundamentally different ecological processes, surveillance systems, and management implications that shape outbreaks in these two groups. We will clarify this choice in the revised manuscript when introducing the separate models. Furthermore, on line 83, we will add “as these two groups are influenced by different ecological processes, surveillance biases, and management contexts”.

I also queried the importance of (dead-end) mammalian infections to a model of the HPAI transmission risk, to which the authors responded: "While spillover events of HPAI into mammals have been documented, these detections are generally considered dead-end infections and do not currently represent sustained transmission chains. As such, they fall outside the scope of our study, which focuses on avian hosts and models ecological suitability for outbreaks in wild and domestic birds." I would argue that any infections, whether they are in dead-end or competent hosts, represent the presence of environmental conditions to support transmission so are certainly relevant to a niche model and therefore within scope. It is certainly understandable if the authors have not been able to access data of mammalian infections, but it is an oversight to dismiss these infections as irrelevant.

We understand the Reviewer’s point, but our study was designed to model HPAI occurrence in avian hosts only. We therefore restricted our analysis to wild birds and domestic poultry, which represent the primary hosts for HPAI circulation and the focus of surveillance and control measures. While mammalian detections have been reported, they are outside the scope of this work.

Correlative ecological niche models, including BRTs, learn relationships between occurrence data and covariate data to make predictions, irrespective of correlations between covariates. I am not convinced that the authors can make any "interpretation" (line 298) that the covariates that are most informative to their models have any "influence" (line 282) on their response variable. Indeed, the observation that "land-use and climatic predictors do not play an important role in the niche ecological models" (line 286), while "intensive chicken population density emerges as a significant predictor" (line 282) begs the question: from an operational perspective, is the best (e.g., most interpretable and quickest to generate) model of HPAI risk a map of poultry farming intensity?

We agree that poultry density may partly reflect reporting bias, but we also assumed it a meaningful predictor of HPAI risk. Its importance in our models is therefore expected. Importantly, our BRT framework does more than reproduce poultry distribution: it captures non-linear relationships and interactions with other covariates, allowing a more nuanced characterisation of risk than a simple poultry density map. Note also that we distinguished in our models intensive and extensive chicken poultry density and duck density. Therefore, it is not a “map of poultry farming intensity”.

At line 282, we used the word “influence” while fully recognising that correlative models cannot establish causality. Indeed, in our analyses, “relative influence” refers to the importance metric produced by the BRT algorithm (Ridgeway, 2020), which measures correlative associations between environmental factors and outbreak occurrences. These scores are interpreted in light of the broader scientific literature, therefore our interpretations build on both our results and existing evidence, rather than on our models alone. However, in the next version of the paper, we will revise the sentence as: “underscoring the strong association of poultry farming practices with HPAI spread (Dhingra et al., 2016)”.

I have more significant concerns about the authors' treatment of sampling bias: "We agree with the Reviewer's comment that poultry density could have potentially been considered to guide the sampling effort of the pseudo-absences to consider when training domestic bird models. We however prefer to keep using a human population density layer as a proxy for surveillance bias to define the relative probability to sample pseudo-absence points in the different pixels of the background area considered when training our ecological niche models. Indeed, given that poultry density is precisely one of the predictors that we aim to test, considering this environmental layer for defining the relative probability to sample pseudo-absences would introduce a certain level of circularity in our analytical procedure, e.g. by artificially increasing to influence of that particular variable in our models." The authors have elected to ignore a fundamental feature of distribution modelling with occurrence-only data: if we include a source of sampling bias as a covariate and do not include it when we sample background data, then that covariate would appear to be correlated with presence. They acknowledge this later in their response to my review: "...assuming a sampling bias correlated with poultry density would result in reducing its effect as a risk factor." In other words, the apparent predictive capacity of poultry density is a function of how the authors have constructed the sampling bias for their models. A reader of the manuscript can reasonably ask the question: to what degree are is the model a model of HPAI transmission risk, and to what degree is the model a model of the observation process? The sentence at lines 474-477 is a helpful addition, however the preceding sentence, "Another approach to sampling pseudo-absences would have been to distribute them according to the density of domestic poultry," (line 474) is included without acknowledgement of the flow-on consequence to one of the key findings of the manuscript, that "...intensive chicken population density emerges as a significant predictor..." (line 282). The additional context on the EMPRES-i dataset at line 475-476 ("the locations of outbreaks ... are often georeferenced using place name nomenclatures") is in conflict with the description of the dataset at line 407 ("precise location coordinates"). Ultimately, the choices that the authors have made are entirely defensible through a clear, concise description of model features and assumptions, and precise language to guide the reader through interpretation of results. I am not satisfied that this is provided in the revised manuscript.

We thank the Reviewer for this important point. To address it, we compared model predictive performance and covariate relative influences obtained when pseudo-absences were weighted by poultry density versus human population density (Author response table 1). The results show that differences between the two approaches are marginal, both in predictive performance (ΔAUC ranging from -0.013 to +0.002) and in the ranking of key predictors (see below Author response images 1 and 2). For instance, intensive chicken density consistently emerged as an important predictor regardless of the bias layer used.

Note: the comparison was conducted using a simplified BRT configuration for computational efficiency (fewer trees, fixed 5-fold random cross-validation, and standardised parameters). Therefore, absolute values of AUC and variable importance may differ slightly from those in the manuscript, but the relative ranking of predictors and the overall conclusions remain consistent.

Given these small differences, we retained the approach using human population density. We agree that poultry density partly reflects surveillance bias as well as true epidemiological risk, and we will clarify this in the revised manuscript by noting that the predictive role of poultry density reflects both biological processes and surveillance systems. Furthermore, on line 289, we will add “We note, however, that intensive poultry density may reflect both surveillance intensity and epidemiological risk, and its predictive role in our models should be interpreted in light of both processes”.

**Author response table 1. sa3table1:** Comparison of model predictive performances (AUC) between pseudo-absence sampling were weighted by poultry density and by human population density across host groups, virus types, and time periods. Differences in AUC values are shown as the value for poultry-weighted minus human-weighted pseudo-absences.

Dataset	AUC (poultry-weighted pseudo-absences)	AUC (human-weighted pseudo-absences)	AUC differences between the two approaches
H5N1, domestic birds, <2020	0.877	0.864	-0.013
H5N1, wild birds, >2020	0.837	0.829	-0.008
H5N1, domestic birds, >2020	0.831	0.828	-0.003
H5Nx, wild birds, <2020	0.856	0.858	0.002
H5Nx, domestic birds, <2020	0.853	0.855	0.002
H5Nx, wild birds, >2020	0.847	0.842	-0.005
H5Nx, domestic birds, >2020	0.835	0.833	-0.002

**Author response image 1. sa3fig1:** Comparison of variable relative influence (%) between models trained with pseudo-absences weighted by poultry density (red) and human population density (blue) for domestic bird outbreaks. Results are shown for four datasets: H5N1 (<2020), H5N1 (>2020), H5Nx (<2020), and H5Nx (>2020).

**Author response image 2. sa3fig2:** Comparison of variable relative influence (%) between models trained with pseudo-absences weighted by poultry density (red) and human population density (blue) for wild bird outbreaks. Results are shown for three datasets: H5N1 (>2020), H5Nx (<2020), and H5Nx (>2020).

The authors have slightly misunderstood my comment on "extrapolation": I referred to "environmental extrapolation" in my review without being particularly explicit about my meaning. By "environmental extrapolation", I meant to ask whether the models were predicting to environments that are outside the extent of environments included in the occurrence data used in the manuscript. The authors appear to have understood this to be a comment on geographic extrapolation, or predicting to areas outside the geographic extent included in occurrence data, e.g.: "For H5Nx post-2020, areas of high predicted ecological suitability, such as Brazil, Bolivia, the Caribbean islands, and Jilin province in China, likely result from extrapolations, as these regions reported few or no outbreaks in the training data" (lines 195-197). Is the model extrapolating in environmental space in these regions? This is unclear. I do not suggest that the authors should carry out further analysis, but the multivariate environmental similarly surface (MESS; see Elith et al., 2010) is a useful tool to visualise environmental extrapolation and aid model interpretation.On the subject of "extrapolation", I am also concerned by the additions at lines 362-370: "...our models extrapolate environmental suitability for H5Nx in wild birds in areas where few or no outbreaks have been reported. This discrepancy may be explained by limited surveillance or underreporting in those regions." The "discrepancy" cited here is a feature of the input dataset, a function of the observation distribution that should be captured in pseudo-absence data. The authors state that Kazakhstan and Central Asia are areas of interest, and that the environments in this region are outside the extent of environments captured in the occurrence dataset, although it is unclear whether "extrapolation" is informed by a quantitative tool like a MESS or judged by some other qualitative test. The authors then cite Australia as an example of a region with some predicted suitability but no HPAI outbreaks to date, however this discussion point is not linked to the idea that the presence of environmental conditions to support transmission need not imply the occurrence of transmission (as in the addition, "...spatial isolation may imply a lower risk of actual occurrences..." at line 214). Ultimately, the authors have not added any clear comment on model uncertainty (e.g., variation between replicated BRTs) as I suggested might be helpful to support their description of model predictions.

Many thanks for the clarification. Indeed, we interpreted your previous comments in terms of geographic extrapolations. We thank the Reviewer for these observations. We will adjust the wording to further clarify that predictions of ecological suitability in areas with few or no reported outbreaks (e.g., Central Asia, Australia) are not model errors but expected extrapolations, since ecological suitability does not imply confirmed transmission (for instance, on Line 362: “our models extrapolate environmental suitability” will be changed to “Interestingly, our models extrapolate geographical”). These predictions indicate potential environments favorable to circulation if the virus were introduced.

In our study, model uncertainty is formally assessed when comparing the predictive performances of our models (Fig. S3, Table S1), the relative influence (Table S3) and response curves (Fig. 2) associated with each environmental factor (Table S2). All the results confirming a good converge between these replicates. Finally, we indeed did not use a quantitative tool such as a MESS to assess extrapolation but did rely on qualitative interpretation of model outputs.

All of my criticisms are, of course, applied with the understanding that niche modelling is imperfect for a disease like HPAI, and that data may be biased/incomplete, etc.: these caveats are common across the niche modelling literature. However, if language around the transmission cycle, the niche, and the interpretation of any of the models is imprecise, which I find it to be in the revised manuscript, it undermines all of the science that is presented in this work.

We respectfully disagree with this comment. The scope of our study and the methods employed are clearly defined in the manuscript, and the limitations of ecological niche modelling in this context are explicitly acknowledged in the Discussion section. While we appreciate the Reviewer’s concern, the comment does not provide specific examples of unclear or imprecise language regarding the transmission cycle, niche, or interpretation of the models. Without such examples, it is difficult to identify further revisions that would improve clarity.

**Reviewer #2 (Public review):**
The geographic range of highly pathogenic avian influenza cases changed substantially around the period 2020, and there is much interest in understanding why. Since 2020 the pathogen irrupted in the Americas and the distribution in Asia changed dramatically. This study aimed to determine which spatial factors (environmental, agronomic and socio-economic) explain the change in numbers and locations of cases reported since 2020 (2020--2023). That's a causal question which they address by applying correlative environmental niche modelling (ENM) approach to the avian influenza case data before (2015--2020) and after 2020 (2020--2023) and separately for confirmed cases in wild and domestic birds. To address their questions they compare the outputs of the respective models, and those of the first global model of the HPAI niche published by Dhingra et al 2016.

We do not agree with this comment. In the manuscript, it is well established that we are quantitatively assessing factors that are associated with occurrences data before and after 2020. We do not claim to determine the causality. One sentence of the Introduction section (lines 75-76) could be confusing, so we intend to modify it in the final revision of our manuscript.

ENM is a correlative approach useful for extrapolating understandings based on sparse geographically referenced observational data over un- or under-sampled areas with similar environmental characteristics in the form of a continuous map. In this case, because the selected covariates about land cover, use, population and environment are broadly available over the entire world, modelled associations between the response and those covariates can be projected (predicted) back to space in the form of a continuous map of the HPAI niche for the entire world.

We fully agree with this assessment of ENM approaches.

Strengths:The authors are clear about expected bias in the detection of cases, such geographic variation in surveillance effort (testing of symptomatic or dead wildlife, testing domestic flocks) and in general more detections near areas of higher human population density (because if a tree falls in a forest and there is no-one there, etc), and take steps to ameliorate those. The authors use boosted regression trees to implement the ENM, which typically feature among the best performing models for this application (also known as habitat suitability models). They ran replicate sets of the analysis for each of their model targets (wild/domestic x pathogen variant), which can help produce stable predictions. Their code and data is provided, though I did not verify that the work was reproducible.The paper can be read as a partial update to the first global model of H5Nx transmission by Dhingra and others published in 2016 and explicitly follows many methodological elements. Because they use the same covariate sets as used by Dhingra et al 2016 (including the comparisons of the performance of the sets in spatial cross-validation) and for both time periods of interest in the current work, comparison of model outputs is possible. The authors further facilitate those comparisons with clear graphics and supplementary analyses and presentation. The models can also be explored interactively at a weblink provided in text, though it would be good to see the model training data there too.The authors' comparison of ENM model outputs generated from the distinct HPAI case datasets is interesting and worthwhile, though for me, only as a response to differently framed research questions.Weaknesses:This well-presented and technically well-executed paper has one major weakness to my mind. I don't believe that ENM models were an appropriate tool to address their stated goal, which was to identify the factors that "explain" changing HPAI epidemiology.Here is how I understand and unpack that weakness:(1) Because of their fundamentally correlative nature, ENMs are not a strong candidate for exploring or inferring causal relationships.(2) Generating ENMs for a species whose distribution is undergoing broad scale range change is complicated and requires particular caution and nuance in interpretation (e.g., Elith et al, 2010, an important general assumption of environmental niche models is that the target species is at some kind of distributional equilibrium at time scales relevant to the model application. In practice that means the species has had an opportunity to reach all suitable habitats and therefore its absence from some can be interpreted as either unfavourable environment or interactions with other species). Here data sets for the response (N5H1 or N5Hx case data in domestic or wild birds) were divided into two periods; 2015--2020, and 2020--2023 based on the rationale that the geographic locations and host-species profile of cases detected in the latter period was suggestive of changed epidemiology. In comparing outputs from multiple ENMs for the same target from distinct time periods the authors are expertly working in, or even dancing around, what is a known grey area, and they need to make the necessary assumptions and caveats obvious to readers.

We thank the Reviewer for this observation. First, we constrained pseudo-absence sampling to countries and regions where outbreaks had been reported, reducing the risk of interpreting non-affected areas as environmentally unsuitable. Second, we deliberately split the outbreak data into two periods (2015-2020 and 2020-2023) because we do not assume a single stable equilibrium across the full study timeframe. This division reflects known epidemiological changes around 2020 and allows each period to be modeled independently. Within each period, ENM outputs are interpreted as associations between outbreaks and covariates, not as equilibrium distributions. Finally, by testing prediction across periods, we assessed both niche stability and potential niche shifts. These clarifications will be added to the manuscript to make our assumptions and limitations explicit.

Line 66, we will add: “Ecological niche model outputs for range-shifting pathogens must therefore be interpreted with caution (Elith et al., 2010). Despite this limitation, correlative ecological niche models remain useful for identifying broad-scale associations and potential shifts in distribution. To account for this, we analysed two distinct time periods (2015-2020 and 2020-2023).”

Line 123, we will revise “These findings underscore the ability of pre-2020 models in forecasting the recent geographic distribution of ecological suitability for H5Nx and H5N1 occurrences” to “These results suggest that pre-2020 models captured broad patterns of suitability for H5Nx and H5N1 outbreaks, while post-2020 models provided a closer fit to the more recent epidemiological situation”.

(3) To generate global prediction maps via ENM, only variables that exist at appropriate resolution over the desired area can be supplied as covariates. What processes could influence changing epidemiology of a pathogen and are their covariates that represent them? Introduction to a new geographic area (continent) with naive population, immunity in previously exposed populations, control measures to limit spread such as vaccination or destruction of vulnerable populations or flocks? Might those control measures be more or less likely depending on the country as a function of its resources and governance? There aren't globally available datasets that speak to those factors, so the question is not why were they omitted but rather was the authors decision to choose ENMs given their question justified? How valuable are insights based on patterns of correlation change when considering different temporal sets of HPAI cases in relation to a common and somewhat anachronistic set of covariates?

We agree that the ecological niche models trained in our study are limited to environmental and host factors, as described in the Methods section with the selection of predictors. While such models cannot capture causality or represent processes such as immunity, control measures, or governance, they remain a useful tool for identifying broad associations between outbreak occurrence and environmental context. Our study cannot infer the full mechanisms driving changes in HPAI epidemiology, but it does provide a globally consistent framework to examine how associations with available covariates vary across time periods.

(4) In general the study is somewhat incoherent with respect to time. Though the case data come from different time periods, each response dataset was modelled separately using exactly the same covariate dataset that predated both sets. That decision should be understood as a strong assumption on the part of the authors that conditions the interpretation: the world (as represented by the covariate set) is immutable, so the model has to return different correlative associations between the case data and the covariates to explain the new data. While the world represented by the selected covariates \*may\* be relatively stable (could be statistically confirmed), what about the world not represented by the covariates (see point 3)?

We used the same covariate layers for both periods, which indeed assumes that these environmental and host factors are relatively stable at the global scale over the short timeframe considered. We believe this assumption is reasonable, as poultry density, land cover, and climate baselines do not change drastically between 2015 and 2023 at the resolution of our analysis. We agree, however, that unmeasured processes such as control measures, immunity, or governance may have changed during this time and are not captured by our covariates.

**Recommendations for the Authors:**

**Reviewer #1 (Recommendations for the authors):**
- Line 400-401: "over the 2003-2016 periods" has an extra "s"; "two host species" (with reference to wild and domestic birds) would be more precise as "two host groups".- Remove comma line 404

Many thanks for these comments, we have modified the text accordingly.

**Reviewer #2 (Recommendations for the authors):**
Most of my work this round is encapsulated in the public part of the review.The authors responded positively to the review efforts from the previous round, but I was underwhelmed with the changes to the text that resulted. Particularly in regard to limiting assumptions - the way that they augmented the text to refer to limitations raised in review downplayed the importance of the assumptions they've made. So they acknowledge the significance of the limitation in their rejoinder, but in the amended text merely note the limitation without giving any sense of what it means for their interpretation of the findings of this study.The abstract and findings are essentially unchanged from the previous draft.I still feel the near causal statements of interpretation about the covariates are concerning. These models really are not a good candidate for supporting the inference that they are making and there seem to be very strong arguments in favour of adding covariates that are not globally available.

We never claimed causal interpretation, and we have consistently framed our analyses in terms of associations rather than mechanisms. We acknowledge that one phrasing in the research questions (“Which factors can explain…”) could be misinterpreted, and we are correcting this in the revised version to read “Which factors are associated with…”. Our approach follows standard ecological niche modelling practice, which identifies statistical associations between occurrence data and covariates. As noted in the Discussion section, these associations should not be interpreted as direct causal mechanisms. Finally, all interpretive points in the manuscript are supported by published literature, and we consider this framing both appropriate and consistent with best practice in ecological niche modelling (ENM) studies.

We assessed predictor contributions using the “relative influence” metric, the terminology reported by the R package “gbm” (Ridgeway, 2020). This metric quantifies the contribution of each variable to model fit across all trees, rescaled to sum to 100%, and should be interpreted as an association rather than a causal effect.

L65-66 The general difficulty of interpreting ENM output with range-shifting species should be cited here to alert readers that they should not blithely attempt what follows at home.I believe that their analysis is interesting and technically very well executed, so it has been a disappointment and hard work to write this assessment. My rough-cut last paragraph of a reframed intro would go something like - there are many reasons in the literature not to do what we are about to do, but here's why we think it can be instructive and informative, within certain guardrails.

To acknowledge this comment and the previous one, we revised lines 65-66 to: “However, recent outbreaks raise questions about whether earlier ecological niche models still accurately predict the current distribution of areas ecologically suitable for the local circulation of HPAI H5 viruses. Ecological niche model outputs for range-shifting pathogens must therefore be interpreted with caution (Elith et al., 2010). Despite this limitation, correlative ecological niche models remain useful for identifying broad-scale associations and potential shifts in distribution.”

We respectfully disagree with the Reviewer’s statement that “there are many reasons in the literature not to do what we are about to do”. All modeling approaches, including mechanistic ones, have limitations, and the literature is clear on both the strengths and constraints of ecological niche models. Our manuscript openly acknowledges these limits and frames our findings accordingly. We therefore believe that our use of an ENM approach is justified and contributes valuable insights within these well-defined boundaries.

Reference: Ridgeway, G. (2007). Generalized Boosted Models: A guide to the gbm package. Update, 1(1), 2007.